# CD34T+ Humanized Mouse Model to Study Mucosal HIV-1 Transmission and Prevention

**DOI:** 10.3390/vaccines9030198

**Published:** 2021-02-27

**Authors:** Kanika Vanshylla, Kathrin Held, Tabea M. Eser, Henning Gruell, Franziska Kleipass, Ricarda Stumpf, Kanika Jain, Daniela Weiland, Jan Münch, Berthold Grüttner, Christof Geldmacher, Florian Klein

**Affiliations:** 1Laboratory of Experimental Immunology, Institute of Virology, Faculty of Medicine and University Hospital Cologne, University of Cologne, 50931 Cologne, Germany; kanika.vanshylla@uk-koeln.de (K.V.); henning.gruell@uk-koeln.de (H.G.); franziska.kleipass@hhu.de (F.K.); ricarda.stumpf@uk-koeln.de (R.S.); kanikajain2006@gmail.com (K.J.); daniela.weiland@uk-koeln.de (D.W.); 2Division of Infectious Diseases and Tropical Medicine, University Hospital, LMU, 80802 Munich, Germany; Kathrin.Held@med.uni-muenchen.de (K.H.); T.Eser@lrz.uni-muenchen.de (T.M.E.); geldmacher@lrz.uni-muenchen.de (C.G.); 3German Center for Infection Research, Partner Site Munich, 80802 Munich, Germany; 4German Center for Infection Research, Partner Site Bonn-Cologne, 50937 Cologne, Germany; 5Institute of Molecular Virology, Ulm University Medical Center, 89081 Ulm, Germany; Jan.Muench@uni-ulm.de; 6Department of Gynecology and Obstetrics, Faculty of Medicine and University Hospital Cologne, University of Cologne, 50931 Cologne, Germany; berthold.gruettner@uk-koeln.de; 7Center for Molecular Medicine Cologne (CMMC), University of Cologne, 50931 Cologne, Germany

**Keywords:** humanized mice, mucosal HIV-1 prevention, broadly neutralizing antibodies

## Abstract

Humanized mice are critical for HIV-1 research, but humanized mice generated from cord blood are inefficient at mucosal HIV-1 transmission. Most mucosal HIV-1 transmission studies in mice require fetal tissue-engraftment, the use of which is highly restricted or prohibited. We present a fetal tissue-independent model called CD34T+ with enhanced human leukocyte levels in the blood and improved T cell homing to the gut-associated lymphoid tissue. CD34T+ mice are highly permissive to intra-rectal HIV-1 infection and also show normal *env* diversification in vivo despite high viral replication. Moreover, mucosal infection in CD34T+ mice can be prevented by infusion of broadly neutralizing antibodies. CD34T+ mice can be rapidly and easily generated using only cord blood cells and do not require any complicated surgical procedures for the humanization process. Therefore, CD34T+ mice provide a novel platform for mucosal HIV-1 transmission studies as well as rapid in vivo testing of novel prevention molecules against HIV-1.

## 1. Introduction

Despite immense progress in the treatment of HIV-1 infection, transmission of the virus continues due to the lack of a vaccine that elicits full protection [1]. Anti-retroviral pre-exposure prophylaxis (PrEP) has shown efficacy in preventing HIV-1 transmission [2]. However, strict daily dosage requirements and disappointing results from clinical trials in African women warrant the need to develop additional alternatives [3]. The pre-clinical in vivo evaluation of the efficacy of anti-HIV-1 molecules has been mostly limited to non-human primates (NHPs) and humanized mice [4,5]. NHPs have a fully functional simian immune system and are susceptible to infection via sexual/mucosal routes making them highly useful for mucosal HIV-1 prevention studies. However, experiments with NHPs require the use of recombinant Simian-HIV to successfully infect primate immune cells [4]. In contrast, humanized mice harbor human immune cells that can be infected with lab-adapted or primary isolates of HIV-1 and have been useful in the evaluation of antiviral efficacy of several antiretroviral drugs and monoclonal antibodies [6,7,8].

Humanized mice are generated by engrafting immune-deficient mouse strains with either adult peripheral blood mononuclear cells (PBMCs), fetal tissue or neonatal cord blood-derived CD34+ hematopoietic stem cells as the primary source of human cells [9]. PBMC-derived humanized mice show rapid expansion of human T cells. However, this is accompanied by acute graft-versus-host disease (GvHD), making this model suitable for short-term HIV-1 studies [10]. Bone marrow-liver-thymus (BLT) mice are generated by surgical implantation of fetus-derived liver and thymus tissue, accompanied by injection of fetal CD34+ hematopoietic stem cells. BLT mice achieve high human cell reconstitution in the gut-associated lymphoid tissue (GALT) and thus have been the primary mouse model for mucosal HIV-1 transmission studies [11,12,13]. However, generating BLT mice requires complicated surgical implantation of human fetal tissue, the use of which is accompanied by both ethical and legal restrictions. Therefore, there is an urgent need to improve existing models or develop new mouse models for studying HIV-1 prevention that do not rely on fetal tissue [14].

A prominent fetal tissue-independent method to humanize mice is to inject cord blood-derived CD34+ human hematopoietic stem cells (HSCs) [9]. CD34+ cell-derived humanized mice (CD34 mice) can be efficiently infected with HIV-1 by intravenous or intraperitoneal application of the virus and are extremely useful for studying the anti-viral potency of new molecules [7,15]. While HIV-1 infection after mucosal challenge has been reported in CD34 mice [16,17], this has not been extendable to all strains of CD34 mice and the low mucosal transmission efficiency in CD34 mice has been attributed to the low human cell reconstitution in the GALT of these mice [18]. 

Since CD34 mice are a widely used and readily available humanized mouse model [19], we aimed to develop this model for generating mice that can also be used for mucosal HIV-1 transmission and prevention. Here, we present a new model for HIV-1 prevention, called CD34T+, that does not rely on fetal tissue but allows robust mucosal HIV-1 transmission. Compared to the established CD34 model, CD34T+ mice achieve high levels of human cell reconstitution in both blood and GALT. This facilitates high rates of mucosal HIV-1 transmission and allows the use of CD34T+ mice as a small animal model for testing anti-HIV-1 molecules for HIV-1 prevention from a relevant route of HIV-1 exposure.

## 2. Methods

### 2.1. Collection of Umbilical Cord Blood and Placenta Derived Cells

Placental and umbilical cord tissue were collected from donors who gave their written consent under a protocol approved by the Institutional Review Board of the University of Cologne under the protocols 16–110 and 18–420. CD34+ HSCs were isolated from the cord blood and perfused placental tissue by using the CD34 Microbead Kit (Miltenyi Biotec, Bergisch Gladbach, Germany) and stored at −150 °C. The mononuclear umbilical cord blood cell (UCBC) fraction was collected as part of the flow-through of the CD34 Microbead kit and stored at −150 °C.

### 2.2. Generation of Humanized Mice

NOD-*Rag1^null^ IL2rg^null^* (NRG) mice were purchased from The Jackson Laboratory and bred and maintained at the Decentralized Animal Facility of the University of Cologne. Mice were housed under specific-pathogen-free conditions with 12-h day/night conditions and given a diet of ssniff food (Ssniff Spezialdiäten GmbH, Soest, Germany). CD34 humanized mice were generated by engraftment with CD34+ HSCs isolated from human umbilical cord blood as follows: 1–5 days old NRG mice were sub-lethally irradiated at a dose of ~2.0 Gy and 4–6 h later injected intra-hepatically with 2 × 10^5^ purified CD34+ HSCs. At 12 weeks of age, peripheral blood from the mice was analyzed to measure human leukocytes and mice harboring at least 1 human CD4+ T cell/μL blood were considered humanized and termed CD34 mice. For generating CD34T+ mice, on day 0, CD34 mice were injected subcutaneously with 250 ng human interleukin-7 (IL-7) (PeproTech, Hamburg, Germany) in PBS (Gibco, Darmstadt, Germany), three hours before intra-peritoneal injection with donor-matched 30–45 × 10^6^ UCBCs derived from the same cord blood donor used for CD34+ HSC engraftment. Mice were administered 250 ng human IL-7 in PBS on day 1, 2, and 7 post UCBC injection. At 2 weeks post UCBC injection, expansion of human leukocytes was analyzed by FACS and all mice were henceforth termed CD34T+ and used for further analysis/experiments without setting a minimum threshold for T cell levels in order to study the entire spectrum of response to UCBC treatment. NRG-UCBC and NRG-PBMC mice were generated by injecting adult NRG mice with 30–40 × 10^6^ UCBCs or human PBMCs and the IL7 injection regimen described above. All mouse experiments were authorized under the protocol AZ.84-02.04.2015.A353 by the State Agency for Nature, Environmental Protection and Consumer Protection North Rhine-Westphalia (LANUV).

### 2.3. Isolation of Cells from Mouse Spleen or Gut-Associated Lymphoid Tissue

Mice were sacrificed using cervical dislocation and the spleen, large intestine, and small intestine were excised. All isolated cells or tissue were stored at 4°C in medium composed of RPMI 1640 medium GlutaMAX^TM^ supplement (Thermo Fisher, Darmstadt, Germany) containing 10% fetal bovine serum (FBS) (Sigma Aldrich, Munich, Germany) and 1% Penicillin/Streptomycin (Gibco) until further processing. The spleen was processed by homogenization of the tissue using a syringe piston on a 70 μm cell strainer (Corning, Kaiserslautern, Germany) and passing the cell suspension through a 18G syringe needle. Erythrocytes were lysed with ACK lysis buffer (Thermo Fisher, Darmstadt, Germany) and the splenic cells were washed in cell medium and analyzed by FACS. For isolation of lymphocytes from gut-associated lymphoid tissue, the intestine was flushed with PBS. 1-cm dissected sections were pre-digested to break down the extracellular matrix by incubating twice at 37 °C for 15 min in medium containing 25 mM EDTA followed by vortexing and passing the solution through a 70 μm cell strainer to obtain the intra-epithelial lymphocyte fraction. The rest of the tissue was digested by incubating twice at 37 °C for 20 min in medium containing 1 mg/mL Collagenase D (Sigma-Aldrich, Munich, Germany), 0.1 mg/mL DnaseI (Sigma-Aldrich, Munich, Germany), and 0.5 mg/mL Dispase (Sigma-Aldrich, Munich, Germany), followed by vortexing and passing the solution through a 70 μm cell strainer (Corning, Kaiserslautern, Germany) to obtain the lamina propria (LP) fraction. The IEL and LP fractions were washed in medium and passed through 70 μm and 40 μm cell strainers consecutively. The lymphocytes were purified by performing a density gradient using 30% (*v*/*v*) Histopaque solution (Sigma-Aldrich, Munich, Germany) and purified cells analyzed by FACS.

### 2.4. Flow Cytometry Analysis of Cell Populations in Mouse Blood or Tissue

Mice were bled from the submandibular vein, the erythrocytes in the blood were lysed with ACK lysis buffer, and the sample stained with the following antibodies: anti-mouse-CD45-PECy7 (BioLegend, Munich, Germany; clone 30-F11), anti-human-CD45-Pacific Orange (Thermo Fisher, Darmstadt, Germany; clone HI30), anti-human-CD19-APC (BD Pharmigen, Heidelberg, Germany; clone HIB19), anti-human-CD3-Pacific Blue (BD Pharmigen, Heidelberg, Germany; clone UCHT1), anti-human-CD4-PE (BD Pharmigen, Heidelberg, Germany; clone L120), anti-human-CD8-FITC (BD Pharmigen, Heidelberg, Germany; clone SK1), anti-human-CD16-AF700 (BD Pharmigen, Heidelberg, Germany; clone CD2F1). The samples were washed and analyzed in FACS Buffer (PBS containing 2% FBS and 2 mM EDTA pH = 8.0). Mouse tissue lymphocytes were isolated as detailed above and 1 × 10^6^ cells were used for FACS staining. FACS data were acquired using a BD FACSAria Fusion flow cytometer and sample analysis was done on the FlowJo v10.2 software with final graphs and statistical analysis being performed in GraphPad Prism 7.

### 2.5. Production of Replication Competent HIV-1 Using 293-T Cells

Replication-competent NL4-3_YU2_ [20] HIV-1 molecular clone was used to transfect Human Embryonic Kidney (HEK) 293T cells using the FuGENE 6 Transfection Reagent (Promega, Walldorf, Germany). The virus culture supernatant was harvested at 48 h post transfection and stored at −80 °C or –150 °C.

### 2.6. Determination of HIV-1 Infectious Titer and bNAb Concentration Using TZM.bl Assay

Determination of Tissue Culture Infectious Dose 50 (TCID50) of HIV-1 culture and plasma antibody concentration by neutralization assay was performed using a luciferase-based TZM.bl assay as previously described [21]. An equimolar tri-mix of 3BNC117, 10-1074 and SF12 was used for generating a standard curve for measuring tri-mix antibody levels from mouse plasma. The plasma was heated to 56 °C for 30 min prior to use in TZM.bl assay for inactivation of complement proteins. Bioluminescence was measured using Luciferin/lysis buffer composed of 10 mM MgCl2, 0.3 mM ATP, 0.5 mM Coenzyme A, 17 mM IGEPAL (all Sigma-Aldrich), and 1 mM D-Luciferin (GoldBio, Lonsee, Germany) in Tris-HCl on a BertholdTech (Berthold Technologies, Bad Wildbad, Germany) TriStar2S luminometer.

### 2.7. HIV-1 Challenge of Mice and Viral Load Measurements

All mouse work with HIV-1 infected mice was carried out under inhalation anaesthesia consisting of 2.5 vol% isoflurane with oxygen as carrier gas. CD34 and CD34T+ mice were challenged on consecutive days intra-rectally with a total of 2.9 × 10^5^ TCID50 NL4-3_YU2_ by pipetting virus at the rectal opening without causing any injury in order to avoid direct systemic infection due to bleeding. For this, mice were anaesthetized, held upside down, and the virus gently pipetted into the rectal opening after which mice were kept in an upside down position for another 2 min to prevent backflow of the virus solution. Mice in the tri-mix group received a single intra-peritoneal injection of 2 mg each of 3BNC117, 10-1074 and SF12 in PBS, 24 h prior to the i.r. challenge. CD34 mice that remained un-infected after the intra-rectal (i.r.) challenge, were challenged with HIV-1 via i.p. injection on consecutive days with a total of 7.3 × 10^5^ TCID50 NL4-3_YU2_. For viral load measurements, HIV-1 RNA was isolated using the QIAcube (Qiagen, Hilden, Germany) from mouse plasma using the QIAamp MinElute Virus Spin Kit (Qiagen, Hilden, Germany) along with a DNaseI (Qiagen, Hilden, Germany) digestion step. Viral loads were determined by quantitative real-time PCR using *gag-*specific primers 6F 5′-CATGTTTTCAGCATTATCAGAAGGA-3′ and 84R 5′-TGCTTGATGTCCCCCCACT-3′ and *gag*-specific probe 56-FAM/CCACCCCACAAGATTTAAACACCATGCTAA/ZenDQ as previously described [8]. qPCR was performed on a Light Cycler 480 II (Roche, Penzberg, Germany) using the TaqMan RNA-to-CT 1-Step Kit (Thermo Fisher, Darmstadt, Germany). An HIV-1 standard, whose copy number was determined using the Cobas 6800 HIV-1 kit (Roche, Penzberg, Germany), was included in every RNA isolation/qPCR run, produced by super infection of SupT1-R5 cells. The limit of quantification of the qPCR was determined to be 384 HIV-1 RNA copies/mL and all values below this limit, including mice that remained completely negative, were assigned values between 100–300 for plotting graphs in GraphPad Prism 7 (San Diego, USA).

### 2.8. Production of bNAbs using 293-6E Cells

3BNC117, 10-1074, and SF12 heavy and light chains were previously cloned into human antibody expression plasmids [22,23,24]. Antibodies were produced by transfection of 293-6E cells (National Research Council Canada) using branched polyethylenimine (PEI) 25kDa (Sigma-Aldrich, Munich, Germany). Cells were maintained at 37 °C and 6% CO_2_ FreeStyle 293 Expression Medium (Thermo Fisher, Darmstadt, Germany) and 0.2% Penicillin/Streptomycin 7 days post transfection, the cell culture supernatant was harvested, filtered with a 0.45 μM Nalgene Rapid Flow filter (Thermo Fisher, Darmstadt, Germany), and incubated overnight at 4 °C with Protein G Sepharose 4 Fast Flow (GE Healthcare, Solingen, Germany) overnight. Antibody bound Sepharose beads were washed on chromatography columns (BioRad, Duesseldorf, Germany) and antibodies were eluted using 0.1M Glycine pH = 3 and immediately buffered in 1M Tris pH = 8. Thereafter, buffer exchange to PBS was performed using 50 kDa Amicon Ultra-15 spin columns (Millipore, Darmstadt, Germany) and the antibodies were sterile filtered using 0.22 μm Ultrafree-CL columns (Millipore, Darmstadt, Germany) and stored at 4 °C.

### 2.9. In-Situ Hybridization for HIV-RNA and Immunofluorescence on Mouse Tissue Samples

Mice were sacrificed using cervical dislocation and the spleen, large intestine (colon), and small intestine (ileum) were excised. Mouse tissues were stored in 30% sucrose after overnight fixation with 4% PFA in PBS (Gibco) until freezing in tissue freezing medium (Leica Biosytems, Nussloch, Germany). 10 µm cryostat tissue sections were cut, mounted on SuperFrost Plus slides (Thermo Scientific, Darmstadt, Germany), and stored at −80 °C until further use. Fluorescence HIV-1 RNA in situ hybridisation was performed using the RNAscope Multiplex Fluorescent Detection Reagent Kit v2 and the RNAscope Probe- V-HIV1-CladeB probe (both ACD, bio-techne, Newark, USA) according to the manufacturer’s instructions with minor modifications. Fixed-frozen tissue sections were baked at 60 °C, re-fixed with 4% PFA in PBS, treated with hydrogen peroxide, and after unmasking of target RNA by heating and protease digestion, probes were hybridised and the signal was amplified. Bound probe was detected with Opal 570 fluorophore (Perkin Elmer, Waltham, USA). Subsequently, the slides were incubated overnight with antibodies against human CD3 (rat IgG1, clone CD3-12, Abcam) and CD4 (rabbit IgG, clone EPR6855, Abcam, ) at 4 °C and bound antibodies were detected by fluorescently labelled anti-rat AlexaFluor 488 and anti-rabbit IgG AlexaFluor 647 antibodies (Thermo Fisher Scientific). Nuclei were counterstained with DAPI (Thermo Fisher Scientific) and slides were embedded in Fluoromount-G (Thermo Fisher Scientific). Immunofluorescence (IF) for human CD3, CD4, and CD8 was carried out as follows: Fixed frozen tissue sections were thawed, re-fixed with 4% PFA in PBS, and after heat induced antigen retrieval with Tris/EDTA at pH 9.0, incubated with antibodies against human CD3 (rat IgG1, clone CD3-12, Abcam), CD4 (rabbit IgG, clone EPR6855, Abcam), and CD8 (mouse IgG1, clone C8/144B, Agilent Dako). Bound primary antibodies were detected by fluorescently labelled anti-mouse IgG AlexaFluor 488, anti-rabbit IgG AlexaFluor 555, and anti-rat AlexaFluor 647 antibodies (Thermo Fisher Scientific). Nuclei were counterstained with DAPI (Thermo Fisher Scientific) and slides were mounted with Fluoromount-G (Thermo Fisher Scientific). Signal was visualized using an inverted Leica DMi8 epifluorescence microscope at the core facility bioimaging of the BMC (LMU) and final image processing was done using ImageJ (NIH, USA) and Adobe Illustrator^®^ (San Jose, CA, USA).

### 2.10. Single Genome Amplification of HIV-1 Env from Mouse Plasma

Plasma RNA was extracted using the QIAamp MinElute Virus Spin Kit (Qiagen, Hilden, Germany) along with a DNaseI (Qiagen, Hilden, Germany) digestion step using the QIAcube (Qiagen, Hilden, Germany). cDNA was generated from plasma RNA using the primer YB383 5′-TTTTTTTTTTTTTTTTTTTTTTTTRAAGCAC-3′ as per the manufacturer’s protocol for the Superscript III Reverse Transcriptase (Thermo Fisher Scientific, Darmstadt, Germany). cDNA was additionally treated with Ribonuclease H (Thermo Fisher) for 20 min at 37 °C and stored at −80 °C. The HIV-1_YU2_
*env* cDNA was amplified by nested PCR using dilutions that would yield <30% positive PCR reactions so that >80% reactions would yield a product derived from a single virus particle. 1^St^ PCR was performed using the primers YB383 5′-TTTTTTTTTTTTTTTTTTTTTTTTRAAGCAC -3′ and YB50 5′-GGCTTAGGCATCTCCTATGGCAGGAAGAA-3′ with the cycling conditions: 98°C for 45 s, 35 cycles of 98 °C for 15 s, 55 °C for 30 s, and 72 °C for 4 min and final amplification at 72 °C for 15 min. 1 μl of the 1st PCR product was used for 2nd PCR with the primers YB49 5′-TAGAAAGAGCAGAAGACAGTGGCAATGA-3′ and YB52 5′-GGTGTGTAGTTCTGCCAATCA GGGAAGWAGCCTTGTG-3′ with the cycling conditions: 98 °C for 45 s, 45 cycles of 98 °C for 15 s, 55 °C for 30 s, and 72 °C for 4 min and final amplification at 72 °C for 15 min. PCR was performed using Phusion Hot Start Flex DNA Polymerase (New England Biolabs, Frankurt am Main, Germany).

### 2.11. Illumina Dye Sequencing of HIV-1 Env Amplicons and Sequence Analysis

NGS library preparation for Illumina Dye sequencing was done as previously described [25]. In brief, the 2nd SGA *env* PCR products were cleaved into approximately 300 bp products by tagmentation using the Nextera DNA Library Prep Kit (Illumina, San Diego, USA). Indices from the Nextera Index Kit (Illumina), followed by adaptors P1 (AATGATACGGCGACCACCGA) and P2 (CAAGCAGAAGACGGCATACGA) were added by limited cycle PCR using the KAPA HiFi Hot Start Ready Mix (Roche, Penzberg, Germany). PCR products were purified using AMPure XP beads (Beckman Coulter, Krefeld, Germany), pooled, and then sequenced using the MiSeq 300-cycle Nano Kit v2 (Illumina) spiked with ~10% PhiX. Paired end reads were assembled as previously described [26] and a consensus sequence was built. Further analysis was done using Geneious R10v10.0.9 where sequences with >75% nucleotide identity across reads were considered. Full-length sequences with high base quality and a maximum of 1 ambiguity were considered for final analysis. Alignments and phylogenetic trees were built using the ClustalOmega v1.2.3 and FastTree v2.1.11 plugins in Geneious R10v10.0.9 (Biomatters, New Zealand).

## 3. Results

### 3.1. CD34T+ Mice have Enhanced Human Leukocyte Levels in the Blood

All umbilical cord blood samples were processed and stored in two fractions: CD34+ hematopoietic stem cells (HSCs) and mononuclear umbilical cord blood cells (UCBCs) (Figure 1A). For generating CD34 mice, newborn NOD-*Rag1^null^ IL2rg^null^* (NRG) mice were irradiated and injected with CD34+ HSCs and analyzed by FACS at 12 weeks of age to confirm the presence of human cells in peripheral blood (Figure 1A). To generate CD34T+ mice, 13–40 weeks old CD34 mice were injected with donor-matched UCBCs and four doses of 250 ng human IL-7, which is reported to help homing of T cells to the gut [27,28] (Figure 1A). We compared CD34T+ mice before (week 0) and after (week 2) UCBC injection, along with a control group of CD34 mice for changes in human leukocyte levels (FACS gating strategy in Appendix A). CD34T+ mice showed increased absolute levels of human CD45+ cells and the primary lymphocyte population to undergo expansion was CD3+ T cells and included both CD4+ and CD8+ T cell populations. In addition, we observed a significant increase in CD16+ cells in CD34T+ mice, while CD19+ B cells and mouse CD45+ leukocytes did not show significant changes after 2 weeks (Figure 1B, Appendix A). In the CD34 control group, levels of all analyzed human and mouse cell populations remained similar over time (Figure 1B, Appendix A). Our analysis encompassed 18 umbilical cord blood donors, confirming that CD34T+ mice can be successfully generated from different donors (Appendix A). These data show that the CD34T+ mice exhibit high T cell reconstitution in the peripheral blood. Next, we also tested non-humanized NRG mice using this treatment regimen and injected either neonatal UCBCs or adult PBMCs, since donor mismatch was not a concern in this case (Appendix A). UCBC reconstitution alone did not result in high levels of peripheral blood T cells (Appendix A). The levels of leukocytes in NRG-PBMC mice were similar to CD34T+ mice, however, NRG-PBMC mice developed very rapid GvHD with symptoms including hunched posture, fur loss, and reduced mobility [29] (Appendix A). Thus, CD34T+ mice were deemed the most appropriate model system for achieving high human cell expansion.

### 3.2. CD34T+ Mice have Enhanced T Cell Homing to GALT

Next, we examined human cell reconstitution in lymphoid tissues by purifying cells from spleen, gut lamina propria (LP), and gut intra-epithelial layer (IEL) (Appendix A). Consistent with the higher human lymphocyte reconstitution in the peripheral blood, CD34T+ mice exhibited significantly higher levels of human CD4+ T cells in the LP and IEL than CD34 mice (Figure 1C, Appendix A). CD34T+ mice also had more CD4+ T cells in the spleen but this difference was not statistically significant (Figure 1C). CD19+ B cell levels in GALT had comparable levels in both CD34 and CD34T+ mice and similar levels of murine CD45+ cells confirmed similar purification efficiency across mice from both groups (Appendix A). To verify that cells analyzed via FACS did not originate from tissue-draining blood vessels, but originated from the mucosal tissue, we performed IF staining of T cell distribution in intact tissue. IF also confirmed the presence of human T cells in gut sections from CD34T+ mice, but none in the sections from CD34 mice (Figure 1D), where FACS analysis of whole gut tissue had also only revealed a scarce human CD4+ T cell population. The majority of human T cells observed in GALT CD34T+ mice were human CD4+ T cells, with few human CD8+ T cells. These T cells were distributed throughout the lamina propria, mostly clustered within the basal layers around crypts but were also present within the lamina propria core of villi (Figure 2E and Appendix A). We did not find any apparent lymphoid structures such as Peyer’s Patches (Figure 1D and Appendix A). In the spleen, typical splenic tissue organization with clear T and B cell zones was not apparent in either mouse model. However, structures resembling periarteriolar lymphoid sheaths (PALS), previously described for humanized NSG mice [30] were observed for some CD34T+ mice (Figure 1D and Figure 2E, and Appendix A). In the spleen as well, human CD4+ T cells formed the majority of the infiltrating human T cell subset. Human CD4+ T cell levels in the blood vs. lymphoid tissues revealed a positive linear relationship with Pearson *r* values of 0.705, 0.782, and 0.827 for LP, IEL, and spleen, respectively (Appendix A). We conclude that the high human CD4+ T cell levels in blood of CD34T+ mice are accompanied by efficient homing of human T cells to the GALT.

### 3.3. CD34T+ Mice can Efficiently Transmit HIV-1 Across the Gut Mucosa

We expected low mucosal HIV-1 transmission in CD34 mice, based on the low human T cell reconstitution in the GALT (Figure 1). As expected, intra-rectal (i.r.) challenge of CD34 mice (Appendix A) with NL4-3_YU2_ HIV-1 resulted in viremia in only 4 out of the 15 challenged mice (Appendix A). Of note, all mice were confirmed to be susceptible to HIV-1 since subsequent intra-peritoneal (i.p.) challenge of the remaining 11 mice resulted in viremia (Appendix A). To identify the tissue reservoir of infected human CD4+ T cells after i.p. infection in CD34 mice, we used RNAscope to visualize HIV-1 RNA in spleen and GALT (ileum and colon) sections. We found infected CD4+ T cells in the spleen, but none in the GALT of the CD34 mice (Appendix A).

In contrast to CD34 mice, in CD34T+ mice, i.r. challenge with the same NL4-3_YU2_ HIV-1 stock all mice became infected, exhibiting 100% success rate (Figure 2A–C). Moreover, mean plasma viremia of ~10^6^ HIV-1 RNA copies/mL at week 3 in CD34T+ mice (Figure 2D) is comparable to early stages of infection in humans [31]. HIV-1 RNAscope staining revealed infected human CD4+T cell reservoirs in both GALT and spleen sections from CD34T+ mice (Figure 2E). The vast majority of HIV-1 RNA positive cells were human CD3+ T cells co-expressing human CD4, with few HIV-1 RNA positive cells only expressing only human CD4 (Figure 2E). In order to analyze the sequence of the viruses that were transmitted across the mucosal membrane, we performed single genome sequence (SGS) analysis of single virions isolated from the plasma of infected mice. SGS and phylogenetic analysis of plasma virions revealed no significant differences in mutational load when comparing i.r. challenge in CD34T+ mice and i.p. challenge in CD34 mice (Figure 3A,B). Thus, CD34T+ mice show similar levels of *env* gene diversification compared to CD34 mice despite the high levels of viral replication (Figure 3C). In summary, these data show that CD34T+ mice efficiently infected following mucosal HIV-1 challenge and can hence be used for mucosal HIV-1 transmission studies.

### 3.4. CD34T+ Mice Can Be Used for HIV-1 Prevention Studies

Our main goal behind developing CD34T+ mice was to have a mucosal HIV-1 prevention model to rapidly evaluate novel anti-HIV-1 molecules, for instance, bNAbs. Antibody-mediated prevention of HIV-1 offers potential advantages like long systemic half-life, limited side-effects, and effector functions that include activation of other immune cells [32]. To test prevention in CD34T+ mice, we used a tri-mix combination of three potent bNAbs, 3BNC117, 10-1074, and SF12 [22,23,24]. These antibodies target different epitopes, namely CD4-binding site, V3 loop and the silent face and show highly potent neutralizing activity against NL4-3_YU2_ HIV-1 (Appendix A). The control group received i.r. HIV-1 challenges, whereas the tri-mix group received an i.p. infusion of the antibodies 1 day before the first i.r. HIV-1 challenge (Figure 4A). As a result, all control group mice became infected within 3 weeks post i.r. HIV-1 challenge (Figure 4B). In contrast, all tri-mix group mice were protected from HIV-1 infection and remained aviremic for up to eight weeks post HIV-1 challenge despite plasma antibody levels dropping to undetectable levels (Figure 4B,C and Appendix A). These results suggest that CD34T+ mice can be used for evaluating the protective function of anti-HIV-1 antibodies.

## 4. Discussion

CD34 mice have most often been used in a therapeutic setting to test the anti-viral potency of new drugs following the establishment of systemic HIV-1 infection [7,8,15]. Several studies on HIV-1 latency [33], antibody effector function [34,35], or pathogen-pathogen interactions [36] have been performed in humanized mice. Advancing humanized mouse models is highly beneficial for examining new therapeutic strategies against infectious diseases and improving our understanding of human immune cell function in vivo [37,38,39,40]. In this study, we present a novel humanized mouse model called CD34T+, which demonstrates increased peripheral and tissue reconstitution with human cells and improved homing of T cells to the GALT. High T cell reconstitution in the GALT creates an ideal tissue environment for mucosal HIV-1 transmission in the CD34T+ mice. We observed 100% infection efficiency in CD34T+ mice when challenged via the rectal mucosa with an infectious molecular strain of HIV-1. Moreover, CD34T+ mice can be utilized as a HIV-1 prevention model as demonstrated by complete inhibition of mucosal infection upon administration of potent bNAbs.

BLT mice exhibit efficient human cell engraftment and have been the primary humanized mouse platform for mucosal HIV-1 prevention studies [11,41,42]. However, a pre-requisite to achieving the high engraftment rates seen in BLT mice is the use of fetal tissue implants. This limits the usage of BLT mice to countries where utilization of fetal tissue is permitted. Cord-blood derived CD34 humanized mice are a widely used mouse model and have been utilized to study various infectious diseases in vivo [9,19]. CD34+ cells from cord blood are more widely accessible and the process of humanization requires simple procedures: sub-lethal irradiation of new born immunodeficient mice, followed by intra-venous or intra-hepatic injection of CD34+ hematopoietic stem cells [43,44]. These attributes make the CD34 mouse model a relatively easy to generate and accessible humanized mouse model. However, mucosal HIV-1 transmission studies with CD34 mice have been rare, since the reconstitution of human cells in the GALT of these mice tends to be low [18]. Notably, some groups have used CD34 mice for mucosal HIV-1 challenge via intra-rectal and intra-vaginal routes, while others have not been successful in infecting these mice via the mucosa [16,17,18]. A possible reason for the ambiguity in ability to infect CD34 mice via mucosal surfaces could be due to differences in engraftment rates of human cells in the different background strains used to generate CD34 mice [45]. In this study, we used mice with the NOD-*Rag1^null^ IL2rg^null^* (NRG) background for generation of our CD34 and CD34T+ humanized mice. CD34T+ mice were generated from CD34 mice ranging in age from 13 weeks up to 40 weeks. While the numbers of UCBCs per donor can be limiting and relative expansion of UCBCs can vary depending on the donor [46], using 18 different placenta donors in this study, we could show that CD34T+ mice can be consistently generated from a diverse set of cord-blood samples. Thus, CD34T+ mice can be rapidly and efficiently generated from the well-established CD34 mice, which we expect to be extendable to immune-deficient mice of different genetic backgrounds. This offers a key advantage to investigators who can thus use their humanized mouse colony for either therapy by conventional i.v. or i.p. challenge routes as well as prevention studies by mucosal challenge without having to switch to a completely new humanized mouse model.

As with every animal model system, there are limitations in the CD34T+ humanized mouse system. We detected onset of anemia in about 30% of animals at 6–8 weeks post UCBC engraftment. However, while HIV-1 therapy studies require up to several months of follow-up to monitor effects on viremia and drug resistance [15], HIV-1 prevention studies are usually performed for short observational windows spanning 4–6 weeks. Therefore, CD34T+ mice are more suited for rather short-term studies, in order to prevent dropouts due to illness. The observed anemia is likely a GvH-related response due to activated T cells derived from the UCBCs. UCBCs are reported to be phenotypically more naive than adult PBMCs and thus our usage of donor matched UCBCs was aimed to minimize the chance of development of GvHD [47,48]. In congruence with this, we observed rapid onset of GvHD in 100% NRG mice within 2 weeks of injection with PBMCs. GvHD is also observed in BLT humanized mice where specific donor HLA class I alleles were shown to increase the incidence of GvHD [49]. Hence, selection of cord blood donor HLA class I alleles with lower probability of GvHD incidence might reduce GvHD occurrences in CD34T+ mice. Additionally, the development of anemia due to human phagocyte mediated erythrocyte depletion has been reported for humanized MISTRG mice [38]. In addition to T cell expansion, CD34T+ mice also show higher levels of human CD16+ cells, which generally comprise monocytes, neutrophils, dendritic cells, and natural killer cells. It is possible that depletion of phagocytes in CD34T+ mice might help to prevent anemia. Despite these limitations, CD34T+ mice are a significant step forward towards our goal of having small animal models to study mucosal HIV-1 transmission and prevention.

Over the last years, bNAbs have emerged as a highly viable option for HIV-1 therapy and prevention [32,50,51,52]. However, VRC01 is the only molecule that has entered human clinical trials for antibody-mediated prevention (NCT02716675 and NCT02568215). It is likely that a globally effective prevention strategy will require a combination of highly potent antibodies [32]. Rapid in vivo testing of such combinations can advance our goal of reducing the global HIV-1 burden. NHPs have been a frequent choice for such testing, but these studies incur high costs, resources, and time, tipping the scale towards the use of small animal models. Moreover, even though all animal research requires ethical considerations, NHPs are highly evolved animals raising specific ethical concerns about their use [53]. Finally, infection of NHPs usually requires the use of recombinant simian-HIV-1 whereas humanized mice can be infected with lab-adapted or primary isolates of HIV-1 for examination of HIV-1 infection and interventions [4]. Although HIV-1 prevention studies in BLT mice have been useful [54], the need for surgical expertise and access to highly restricted human fetal tissue complicates their large-scale use. The need for alternative models is highlighted by the recent notice from the National Institute of Health (NOT-OD-19-042) calling for development of fetal tissue-independent humanized mice. CD34T+ mice can be generated without complex surgery, are fetal tissue-independent, and still achieve high human cell reconstitution. These attributes, along with the high efficiency of mucosal HIV-1 transmission, make CD34T+ mice a useful pre-clinical animal model for testing of newly designed antibodies/drugs for HIV-1 prevention.

## Figures and Tables

**Figure 1 vaccines-09-00198-f001:**
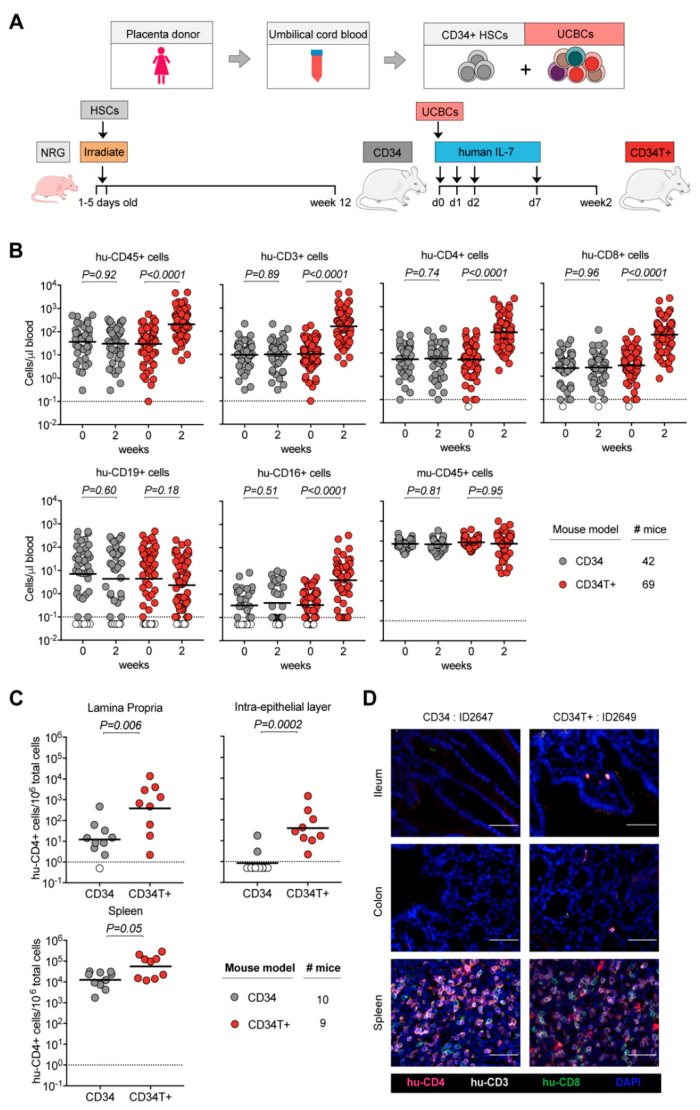
Generation of CD34T+ mice with enhanced reconstitution levels of human cells. (**A**) Human umbilical cord blood from each donor was processed and stored as CD34+ HSCs and UCBCs. 1–5 days old NRG mice were subjected to sub-lethal irradiation and intra-hepatic injection of CD34+ HSCs. At 12 weeks of age, mice that had positively engrafted human leukocytes were termed CD34 mice. To obtain CD34T+ mice, CD34 mice were injected with donor-matched UCBCs and human-IL-7 at the indicated time-points. Two weeks post UCBC injection, the resulting mice were termed CD34T+. (**B**) FACS analysis of leukocyte populations in the blood of CD34T+ mice before UCBC injection (week 0) and 2 weeks after UCBC injection (week 2) along with untreated CD34 mice analyzed in parallel. Points shown in white were below limit of detection of assay. Statistical analysis done using Wilcoxon matched-pairs signed rank test (**C**) FACS analysis of CD34 and CD34T+ mice for the presence of human CD4+ T cells in the gut LP, IEL and the spleen at week 2. Points shown in white were below limit of detection of assay. Statistical analysis was done using Mann-Whitney U test. (**D**) IF analysis of human T cells in the ileum, colon and spleen of CD34 and CD34T+ mice at week 2. Human CD3+ (white), CD4+ (red), CD8+ (green) cells were labeled for detection along with DAPI for visualizing the nucleus. Scale bar is 50 μm. Three mice per group were analyzed with at least three different slices from different areas of each mouse tissue being analyzed. Representative images were selected for display.

**Figure 2 vaccines-09-00198-f002:**
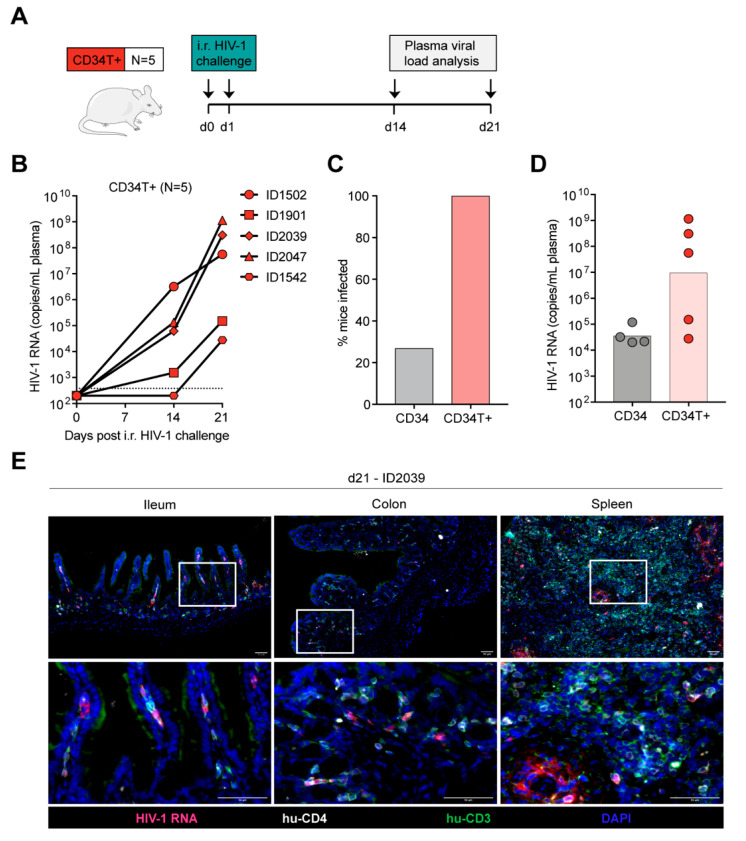
Efficient mucosal transmission of HIV-1 in CD34T+ mice. (**A**) Schematic representation of the experiment design. CD34T+ control mice were given i.r. challenges with NL4-3_YU2_ HIV-1 on consecutive days and bled at the indicated time points for viral load measurements. (**B**) Plasma HIV-1 RNA levels in the blood of CD34T+ mice after intra-rectal challenge with NL4-3_YU2_ HIV-1. The dotted line represents the quantification limit of 384 HIV-1 RNA copies/mL of the assay. (**C**) Comparison of infectivity rates (in %) of CD34 and CD34T+ mice post i.r. HIV-1 challenge. (**D**) Plasma HIV-1 RNA levels at week 3 after i.r. challenge with HIV-1 of CD34 and CD34T+ mice. (**E**) RNA-scope-based detection of HIV-1 infected cells in the ileum, colon and spleen of CD34T+ mice harvested at day 21 post i.r. HIV-1 challenge. HIV-1 RNA (pink), CD4+ cells (white), CD3+ cells (green) were labeled by in-situ hybridization and IF and nuclei were stained with DAPI. Scale bar is 50 μm. 3 mice per group and 3 tissue sections per mouse were analyzed. Representative images were selected for display. i.r. intra-rectal.

**Figure 3 vaccines-09-00198-f003:**
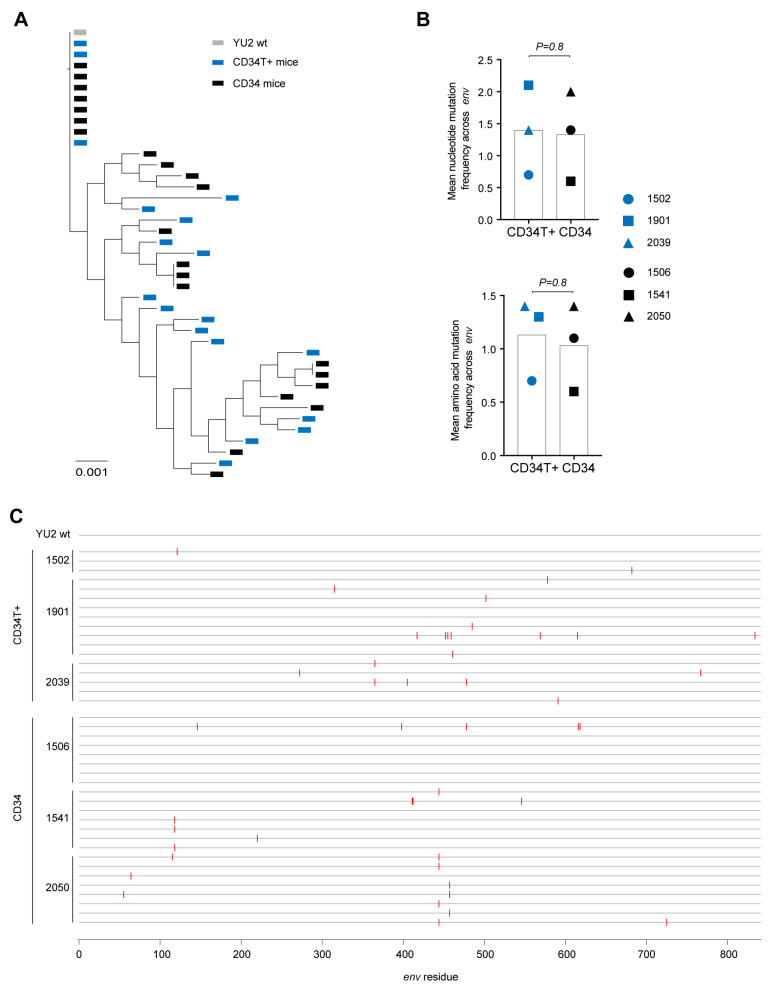
In vivo diversification and evolution of HIV-1 infected mice. (**A**) Phylogenetic tree of SGS-derived *env* sequences from the plasma of CD34T+ mice and CD34 mice at day 21 post post i.r. or i.p. challenge with NL4-3_YU2_ HIV-1 respectively. (**B**) Analysis of mutation rates at the amino acid and nucleotide level in the *env* gene of mice from (**A**). (**C**) Amino acid alignment of plasma SGS-derived *env* sequences from mice analyzed in (**A**,**B**). Red bars indicate amino acid residues with mutations relative to the wild-type NL4-3_YU2_ HIV-1 *env*. Amino acid numbering is based on HIV-1_YU2_
*env*. i.r. intra-rectal, i.p. intra-peritoneal.

**Figure 4 vaccines-09-00198-f004:**
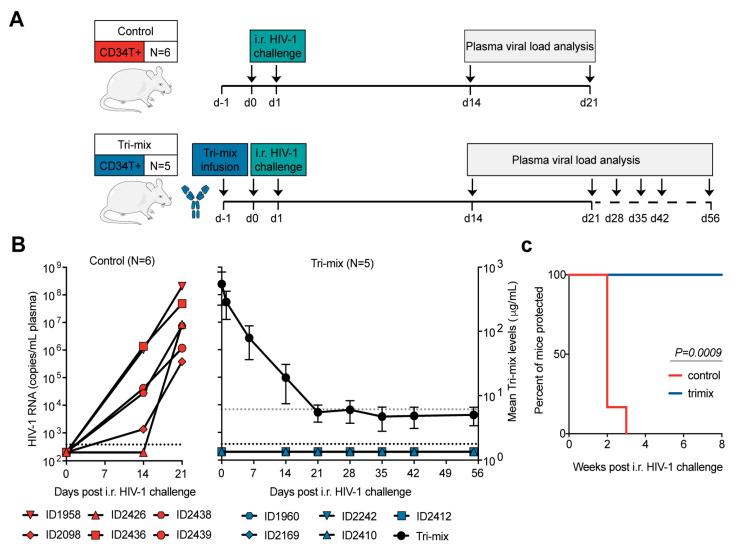
Antibody-mediated prevention of mucosal HIV-1 infection in CD34T+ mice. (**A**) Schematic representation of the experimental design. CD34T+ control mice were given high dose i.r. challenges with NL4-3_YU2_ HIV-1 on consecutive days, whereas, the CD34T+ tri-mix mice additionally received i.p. injection of 2 mg each of 3BNC117, 10-1074 and SF12 one day before the 1st HIV-1 challenge. (**B**) Plasma HIV-1 RNA levels over time in the control and tri-mix mice. The lower black dotted line represents the assay quantification limit of 384 HIV-1 RNA copies/mL plasma. The right *Y*-axis for the tri-mix mice shows mean *(N=* 5) neutralizing antibody tri-mix levels in plasma over time. The upper grey dotted line represents the limit of detection of the antibody measurement assay. (**C**) Kaplan-Meier curve depicting rate of protection from mucosal HIV-1 infection in control and tri-mix mice from (**B**) statistical analysis done with Log-rank (Mantel-Cox) test. i.r. intra-rectal.

## Data Availability

All flow cytometry, viral load measurement and neutralization assay data has been included in main figures and supplement. All SGS-derived HIV-1 envelope sequences have been deposited at Genbank (accession numbers 660543-660582). All original unprocessed imaging files will be made available upon request.

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
