# Peer review of "CD34T+ Humanized Mouse Model to Study Mucosal HIV-1 Transmission and Prevention"

_vaccines, 2021, doi:10.3390/vaccines9030198_

Round 1

Reviewer 1 Report

The authors have done a great piece of work to create a novel platform to study mucosal HIV transmission in the article titled "CD34T+ humanized mouse model to study mucosal HIV-1 transmission and prevention." Although the article is well written and scientifically sound, there are few concerns before it gets published. Those are as follows:

It would be good if authors can provide Institutional Animal Care and Use Committee (IACUC) and Institutional Review Board approval number.

Many abbreviations are used without explaining them, like inline no 84 "UCBCs" and line no.154 "i.r."

Results:

Compared to non-humanized NRG mice, UCBC reconstitution does not generate any change, but PBMC was similar to CD34+ mice. Although the mCD45+ cells were higher in these mice, what was the maximum time interval authors checked the models' leukocytes?

How long these mice survived?

Results seem fascinating that CD34+ mice can efficiently transmit HIV-1 across the gut. Have authors checked the change in leukocytes proportion after the infection?

Discussion:

It would be good if authors can give more details about the GvHD in other models than the proposed CD34+ model.

Overall it is a very well written manuscript with valid and scientifically appropriate experiments. These results could help extend and elaborate future studies in Humanized mouse models to study different viral transmissions.

Reviewer 2 Report

The manuscript describes a promising preclinical animal model of mucosal HIV-1 infection. The authors have generated the so-called CD34T+ humanised mice, through the use of CD34+ cells isolated from cord blood and the subsequent application of the mononuclear cell fraction from the cord blood in combination with hIL-17. Since this model is independent of fetal tissue engraftment which provides many advantages. 

The mouse model is very interesting. However, I would consider it as very beneficial if the authors could expand their phenotyping approach to provide a full picture of the CD34T+ mice. This would definitely be of use for any future studies using these mice.

The work is well described, the results are presented very well and the manuscript is well written.

There are a few areas that the authors could expand on. One would be the complete phenotyping of the animals like it has recently been done on CD34 mice (Blumich et al., 2021. Vet Pathol. 58(1):161-180). In the present study, the authors only looked at spleen and intestines. However, in CD34 mice, the human cells distribute substantially more widely. It would be very useful to check whether this also applies to the CD34T+ mice. Considering that the animals show substantially more hCD16+ cells, in particular the bone marrow would be of interest. Also, CD34 mice often develop changes that are consistent with GvHD; do these develop to the same extent in CD34+ mice and if so, when? Do they look similar or is the composition of inflammatory infiltrates different? The authors mention that they “detected onset of anemia in some animals”. This information is not provided in the Results section. Is there a specific reason for this? Considering the extent to which anaemia is discussed, it would be advisable to include the relevant results.

There are some specific issues that should be addressed. These are listed in a consecutive manner.   

Introduction

Line 47: Please re-place the word “several” (behind “of”).

Line 72: Please delete “use”.

Line 73: Please reword “HIV-1 prevention” in order to make clear that you are talking about prevention of infection.

Line 75: Please delete “the” in front of “GALT”.

Materials and Methods

Line 89: Please provide information on the supplier of the food.

Line 98/99: The process “expansion of human leukocytes was analyzed by FACS” should be explained. Was there a threshold for the determination of CD34T+ mice?

Lines 100 and 101: It should be mentioned from which species the PBMCs originate.

Line 112: Delete “tissue”.

Lines 124 to 128: The correct nomenclature for the antibodies would be “anti-mouse CD45-PECy7” and so on. The way it is currently written this is a bit of lab slang…

Line 153: Apologies if I overlooked this, but was “the tri-mix group” defined? If not, the text should maybe be a bit more elaborate at this point.

Line 179: As it appears the author examined the spleen and intestines by RNA-ISH for HIV-1 and by IF for CD3, CD4 and CD8, but did not really perform a histological examination (assessment of histological features based on an HE-stained section, for example). The heading should be amended accordingly.

Line 188: Ii is a bit confusing to speak about “antigen-retrieval” in the context of RNA-ISH. Please reword.

Line 190: Please state which of the primary antibodies was made in rat and which in rabbit.

Line 194: Replace “Immunohistochemistry” by “immunofluorescence”.

Line 195: Which type of antigen retrieval was applied for the different protocols?

Line 196: Again, please state which of the primary antibodies was made in rat and which in rabbit and mouse.

Line 199: I guess the authors used Fluoromount-G to coverslip the slides. They certainly did not “embed” the slides in it. Please reword.

Results

Lines 239: I suggest to delete “and Methods”. Does it make sense to refer to “Methods” in the results chapter? It is obvious that a manuscript also contains a methods sections that appropriately explains the procedures.

Line 248: Replace “change” by “changes”.

Line 256: It should be explained on which basis the authors claim that “NRG-PBMC mice developed very rapid GvHD” (both in the text and in the legend of Supplementary Fig. 2B) as this is not necessarily obvious to the reader. An appropriate reference should also be included to explain why GvHD can be diagnosed on the basis of the examined parameters.

Line 259: If it is indeed only T cells (or specifically only CD4+ T cells) that are found to a higher extent in the GALT of the CD34T+ mice, the heading should be reworded.

Line 263: The authors only mention CD4+ cells now. What about the all the other cell (lymphocyte) populations they tested for (lines 124-128)? Since these were apparently tested for, the results should be provided in some form.

Line 267: Replace “Immunohistochemistry” by “immunofluorescence” and, if an abbreviation is wanted, “IHC” by “IF”. The same comment applies to the legend of Figure 1D.

Line 268: This statement (“confirmed the presence of human T cells in gut sections from CD34T+ mice”) is a bit too brief. Firstly, it should be commented on the CD4+ and CD8+ T cells shown in Fig. 1D, and it should be commented on the expression in the spleen. It would also the useful to comment on the distribution of the T cells in the spleen in the CD34T+ mice, since it has recently been shown that in CD34 humanised mice the human cells clustered around splenic arterioles and formed PALS-like structures (Blumich et al., 2021. Vet Pathol. 58(1):161-180).  

Line 283/284: The information provided should be more detailed with regard to the type of “infected cells”, since the magnification in Fig S4D does not allow to identify the types of cells that are HIV-1 RNA positive.  

Line 296/297: I would expect this final sentence and strong statement in the discussion but not after a results paragraph.

Line 304: Replace “if” by “of”.

Line 314/315: It might be good to tone down the final sentence a slight bit to “The results suggest that CD34T+ mice can….”.

Line 367: Is CD16 not expressed also by neutrophils? These are the most effective professional phagocytes, so might be worth mentioning.

Line 376-377: When speaking about NHPs as the “dominant choice” (maybe not the best wording) of the testing of potent antibodies and the attempts to replace these by small animal models, it might be worth mentioning the ethical issues often raised with regard to the use of NHP, and not restrict the argument to money and time.

Line 384: It might be preferable to replace “platform” by “animal model”.   

Discussion

Line 325: Maybe replace “high” by “increased”?

Line 326: Maybe replace “makes” by “creates”?

Line 335: Would it not be better to refer to countries instead of “regions”?

Line 348: Delete “till” and “old”.

Line 349/350: Would the statement in the first part of the sentence not require a reference?
